# Evidence for adaptive evolution in the receptor-binding domain of seasonal coronaviruses OC43 and 229e

Kathryn E Kistler[1,2]*, Trevor Bedford[1,2]

[1]Molecular and Cellular Biology Program, University of Washington, Seattle, United States; [2]Vaccine and Infectious Disease Division, Fred Hutchinson Cancer Research Center, Seattle, United States

**Abstract** Seasonal coronaviruses (OC43, 229E, NL63, and HKU1) are endemic to the human population, regularly infecting and reinfecting humans while typically causing asymptomatic to mild respiratory infections. It is not known to what extent reinfection by these viruses is due to waning immune memory or antigenic drift of the viruses. Here we address the influence of antigenic drift on immune evasion of seasonal coronaviruses. We provide evidence that at least two of these viruses, OC43 and 229E, are undergoing adaptive evolution in regions of the viral spike protein that are exposed to human humoral immunity. This suggests that reinfection may be due, in part, to positively selected genetic changes in these viruses that enable them to escape recognition by the immune system. It is possible that, as with seasonal influenza, these adaptive changes in antigenic regions of the virus would necessitate continual reformulation of a vaccine made against them.

## Introduction

Coronaviruses were first identified in the 1960s and, in the decades that followed, human coronaviruses (HCoVs) received a considerable amount of attention in the field of infectious disease research. At this time, two species of HCoV, OC43 and 229E, were identified as the causative agents of roughly 15% of common colds (*McIntosh, 1974*; *Heikkinen and Järvinen, 2003*). Infections with these viruses were shown to exhibit seasonal patterns, peaking in January to March in the Northern Hemisphere, as well as yearly variation, with the greatest incidence occurring every 2–4 years (*Monto and Lim, 1974*; *Hamre and Beem, 1972*). Subsequently, two additional seasonal HCoVs, HKU1 and NL63, have entered the human population. These four HCoVs endemic to the human population usually cause mild respiratory infections, but occasionally result in more severe disease in immunocompromised patients or the elderly (*Liu et al., 2020b*). In the past 20 years, three additional HCoVs (SARS-CoV-1, MERS-CoV, and SARS-CoV-2) have emerged, which cause more severe respiratory illness. At the writing of this paper, amidst the SARS-CoV-2 pandemic, no vaccine for any HCoV is currently available, though many candidate SARS-CoV-2 vaccines are in production and clinical trials (*Krammer, 2020*).

Coronaviruses are named for the ray-like projections of spike protein that decorate their surface. Inside these virions is a positive-sense RNA genome of roughly 30 kB (*Li, 2016*). This large genome size can accommodate more genetic variation than a smaller genome (*Woo et al., 2009*). Genome flexibility, coupled with a RNA virus error-prone polymerase (*Drake, 1993*) and a high rate of homologous recombination (*Pasternak et al., 2006*), creates genetic diversity that is acted upon by evolutionary pressures that select for viral replication. This spawns much of the diversity within and between coronaviruses species (*Woo et al., 2009*; *Hon et al., 2008*) and can contribute to the virus'

*For correspondence:
kistlerk@uw.edu

Competing interests: The authors declare that no competing interests exist.

ability to jump species-barriers, allowing a previously zoonotic CoV to infect and replicate in humans.

The battle between virus and host results in selective pressure for mutations that alter viral antigens in a way that evades immune recognition. Antigenic evolution, or antigenic drift, leaves a characteristic mark of positively selected epitopes within the viral proteins most exposed to the host immune system (*Smith et al., 2004*). For CoVs, this is the spike protein, exposed on the surface of the virion to human humoral immunity. Some human respiratory illnesses caused by RNA viruses, like seasonal influenza (*Smith et al., 2004*), evolve antigenically, while others, like measles, do not (*Fulton et al., 2015*). Because of this, seasonal influenza vaccines must be reformulated on a nearly annual basis, while measles vaccines typically provide lifelong protection. Whether HCoVs undergo antigenic drift is relevant not only to understanding HCoV evolution and natural immunity against HCoVs, but also to predicting the duration of a vaccine's effectiveness.

Early evidence that closely related HCoVs are antigenically diverse comes from a 1980s human challenge study in which subjects were infected and then reinfected with a variety of 229E-related strains (*Reed, 1984*). All subjects developed symptoms and shed virus upon initial virus inoculation. After about a year, subjects who were re-inoculated with the same strain did not show symptoms or shed virus. However, the majority of subjects who were re-inoculated with a heterologous strain developed symptoms and shed virus. This suggests that immunity mounted against 229E viruses provides protection against some, but not all, other 229E strains. This is a result that would be expected of an antigenically evolving virus.

More recent studies have identified eight OC43 genotypes and, in East Asian populations, certain genotypes were shown to temporally replace other genotypes (*Lau et al., 2011*; *Zhang et al., 2015*; *Zhu et al., 2018*). Whether certain genotypes predominate due to antigenic differences that confer a fitness advantage is not known. However, evidence for selection in the spike protein of one of these dominant OC43 genotypes has been provided by *dN/dS*, a standard computational method for detecting positive selection (*Ren et al., 2015*). This method has also been used to suggest positive selection in the spike protein of 229E (*Chibo and Birch, 2006*). Additionally, two genetically distinct groupings (each of which include multiple of the aforementioned eight genotypes) of OC43 viruses have been shown to alternate in prevalence within a Japanese community, meaning that the majority of OC43 infections are caused by one group for about 2–4 years at which point the other group begins to account for the bulk of infections. It has been suggested that antigenic differences between these groups contribute to this epidemic switching (*Komabayashi et al., 2020*).

However, a similar surveillance of the NL63 genotypes circulating in Kilifi, Kenya found that NL63 genotypes persist for relatively long periods of time, that people become reinfected by the same genotype, and that reinfections are often enhanced by prior infection (*Kiyuka et al., 2018*). These findings are inconsistent with antigenic evolution in NL63.

Here we use a variety of computational approaches to detect adaptive evolution in spike and comparator proteins in HCoVs. These methods were designed as improvements to *dN/dS* with the intention of identifying adaptive substitutions within a serially sampled RNA virus population. We focus on the seasonal HCoVs that have been continually circulating in humans: OC43, 229E, HKU1, and NL63. Our analyses of nonsynonymous divergence, rate of adaptive substitutions, and Time to Most Recent Common Ancestor (TMRCA) provide evidence that the spike protein of OC43 and 229E is under positive selection. Though we conduct these analyses on HKU1 and NL63, we do not observe evidence for adaptive evolution in the spike protein of these viruses. For HKU1, there is not enough longitudinal sequencing data available for us to confidently make conclusions as to whether or not this lack of evidence reflects an actual lack of adaptive evolution.

## Results

### Phylogenetic consideration of viral diversity and recombination in OC43 and 229e

We constructed time-resolved phylogenies of the OC43 and 229E using publicly accessible sequenced isolates. A cursory look at these trees confirms previous reports that substantial diversity exists within each viral species (*Zhang et al., 2015*; *Komabayashi et al., 2020*; *Lau et al., 2011*). Additionally, the trees form ladder-like topologies with isolate tips arranged into temporal clusters

rather than geographic clusters, indicating a single global population rather than geographically isolated populations of virus. The phylogeny of OC43 bifurcates immediately from the root (*Figure 1*), indicating that OC43 consists of multiple, co-evolving lineages. Because of the distinct evolutionary histories, it is appropriate to conduct phylogenetic analyses separately for each lineage. We have arbitrarily labeled these lineages 'A' and 'B' (*Figure 1*).

Because recombination is common among coronaviruses (*Pasternak et al., 2006*; *Hon et al., 2008*; *Lau et al., 2011*), we built separate phylogenies for each viral gene. In the absence of recombination, each tree should show the same evolutionary relationships between viral isolates. A dramatic difference in a given isolate's position on one tree versus another is strongly indicative of recombination (*Kosakovsky Pond et al., 2006*). Comparing the RNA-dependent RNA polymerase (RdRp) and spike trees reveals this pattern of recombination in some isolates (*Figure 1—figure supplement 1A*). A comparison of the trees of the S1 and S2 sub-domains of spike shows more limited evidence for intragenic recombination (*Figure 1—figure supplement 1B*), which is consistent with the fact that the distance between two genetic loci is inversely related to the chance that these loci remain linked during a recombination event. Though intragenic recombination likely does occur occasionally, analyzing genes, rather than isolates, greatly reduces the contribution of recombination to genetic variation in our analyses.

Thus, in all of our analyses, we use alignments and phylogenies of sequences of single genes (or genomic regions) rather than whole genome sequences of isolates. We designate the lineage of those genes (or genomic regions) based on the gene's phylogeny. Though most isolates contain all genes from the same lineage, some isolates have, say, a lineage A spike gene and a lineage B RdRp gene. This strategy allows us to consider the evolution of each gene separately and interrogate the selective pressures acting on them.

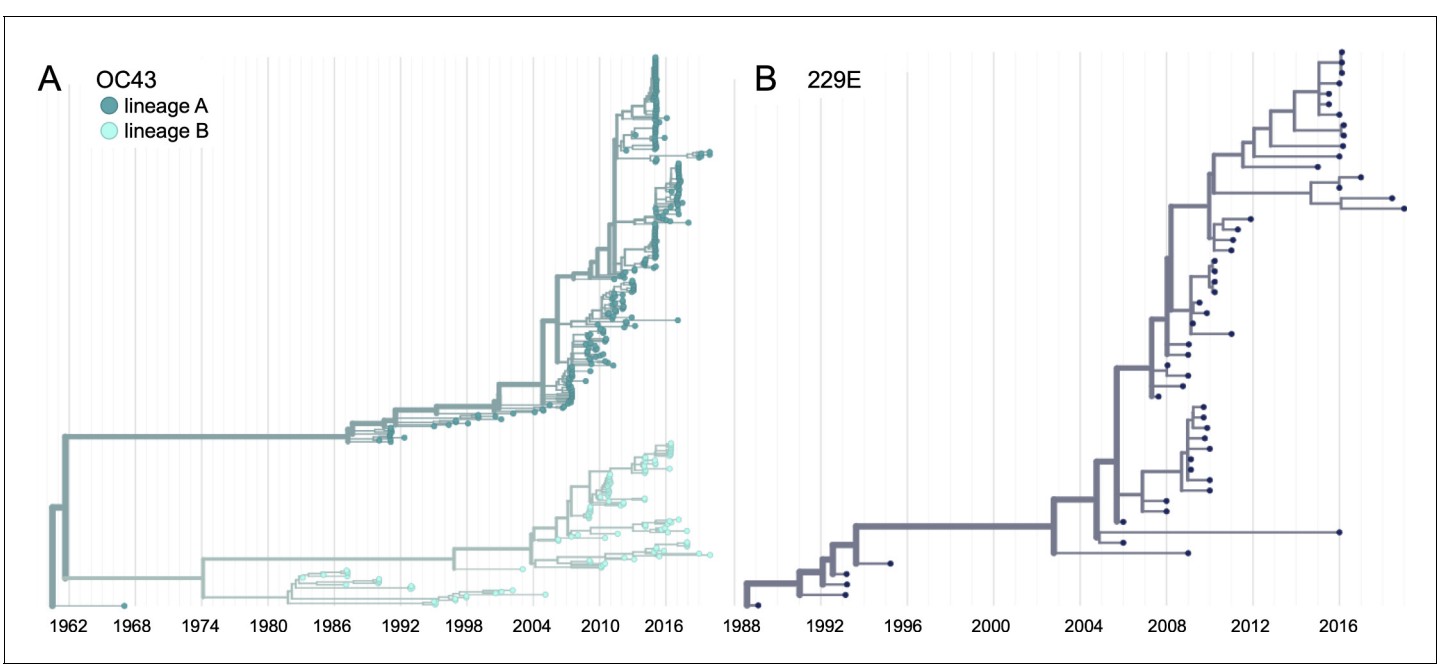

**Figure 1.** Phylogenetic trees for spike gene of seasonal human coronaviruses (HCoVs) OC43 and 229E. Phylogenies built from (**A**) OC43 spike sequences from 389 isolates over 53 years, and (**B**) 229E spike sequences from 54 isolates over 31 years. OC43 bifurcates immediately after the root and is split into two lineages: lineage A (dark teal) and lineage B (light teal). 229E contains just one lineage (dark blue). For the analyses in this paper, the evolution of each gene (or genomic region) is considered separately, so phylogenies are built for each viral gene, and those phylogenies are used to split isolates into lineages for each gene. These are temporally resolved phylogenies with year shown on the x-axis. The clock rate estimate is $5 \times 10^{-4}$ substitutions per nucleotide site per year for OC43 and $6 \times 10^{-4}$ for 229E.

The online version of this article includes the following figure supplement(s) for figure 1:

**Figure supplement 1.** Recombination occurs between human coronavirus (HCoV) isolates.
**Figure supplement 2.** Phylogenetic trees for seasonal human coronaviruses (HCoVs) NL63 and HKU1.

It is worth noting that the analyses we use here to detect adaptive evolution canonically presume that selective pressures are acting on single nucleotide polymorphisms (SNPs). However, it is possible that recombination also contributes to the genetic variation that is acted on by immune selection. This would be most likely to occur if two closely related genomes recombine, resulting in the introduction of a small amount of genetic diversity without disrupting crucial functions. Our analyses do not aim to determine the source of genetic variation (i.e. SNPs or recombination), but rather focus on identifying if and how selection acts on this variation.

Because of its essential role in viral replication and lack of antibody exposure, we expect RdRp to be under purifying selection to maintain its structure and function. If HCoVs evolve antigenically, we expect to see adaptive evolution in spike, and particularly in the S1 domain of spike (*Hofmann et al., 2006*; *Hulswit et al., 2019*), due to its exposed location at the virion's surface and interaction with the host receptor. Mutations that escape from population immunity are beneficial to the virus and so are driven to fixation by positive selection. This results in adaptive evolution of the virus population.

## Phylogenetic inference of substitution prevalence within spike

Using phylogenies constructed from the spike gene, we tallied the number of independent amino acid substitutions at each position within spike. The average number of substitutions per site is higher in S1 than S2 for HCoV lineages in OC43 and 229E (*Figure 2A*). We focus on S1 rather than the receptor-binding domain (RBD) within S1 in our analyses, because it is known that neutralizing antibodies bind to epitopes within the N-terminal domain (NTD) as well as the RBD of S1 (*Liu et al., 2020a*; *Zhang et al., 2018*; *Zhou et al., 2019*). A greater occurrence of repeated substitutions is

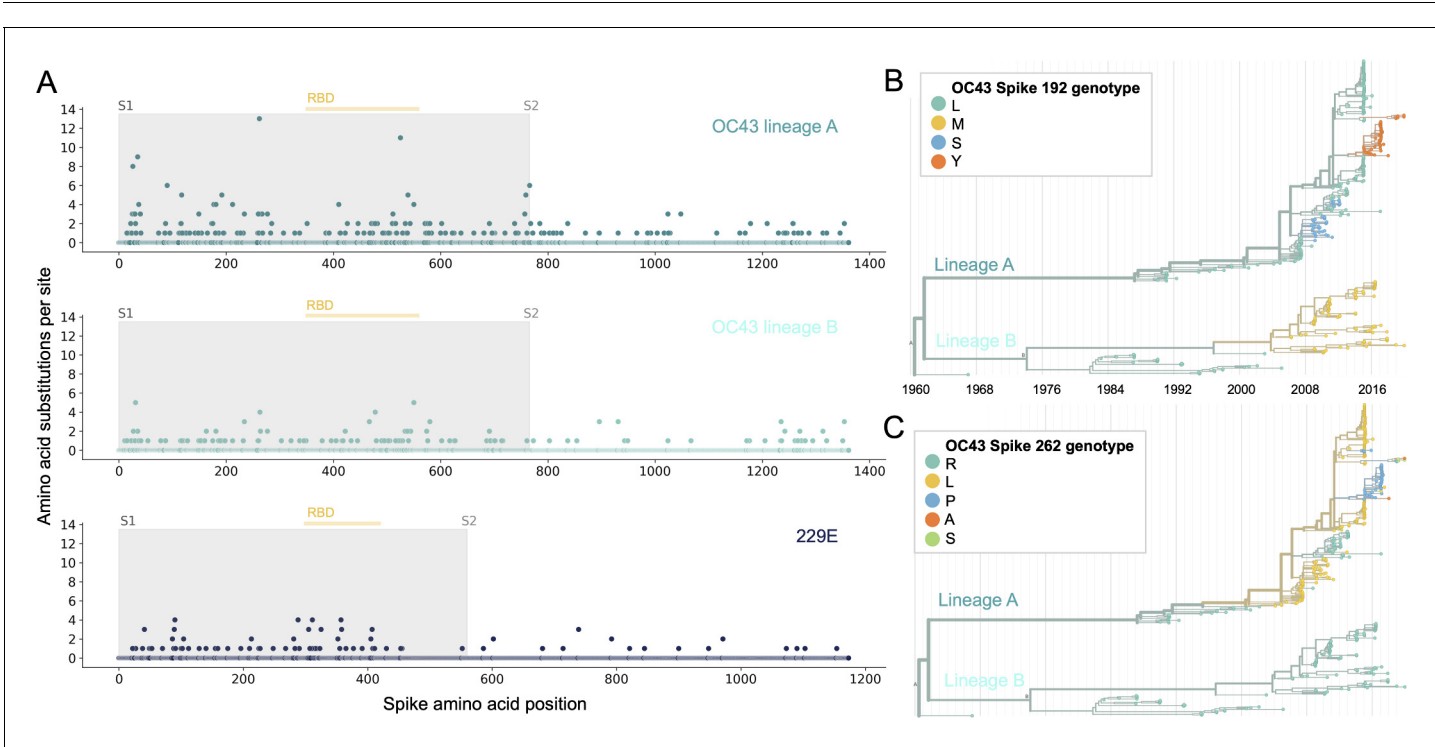

**Figure 2.** More sites mutate repeatedly within spike S1 versus S2. (A) Number of substitutions observed at each amino acid position in the spike gene throughout the phylogeny. S1 (gray) and S2 (white) are indicated by shading and the number of substitutions per site is indicated by a dot and color-coded by human coronavirus (HCoV) lineage. The putative receptor-binding domains for 229E (*Li et al., 2019*) and the putative domain for OC43 (*Lau et al., 2011*) are indicated with light yellow bars. Asterisks indicate two example positions (192 and 262), which mutate repeatedly throughout the OC43 lineage A phylogeny. The OC43 phylogeny built from spike sequences and color-coded by genotype at positions 192 and 262 is shown in (B) and (C), respectively.

The online version of this article includes the following figure supplement(s) for figure 2:

**Figure supplement 1.** Mutations at each position within Spike for NL63 and HKU1.

expected if some mutations within S1 confer immune avoidance. Alternatively, these repeated substitutions could be a result of high mutation rate and random genetic drift as has been shown at particular types of sites in SARS-CoV-2 (*van Dorp et al., 2020*). However, this latter hypothesis should affect all regions of the genome equally and should not result in a greater number of repeated substitutions in S1 than S2.

If the repeated mutations are a product of immune selection, not only should S1 contain more repeated mutations, but we would also expect these mutations to spread widely after they occur due to their selective advantage. Additionally, we expect sites within S1 to experience diversifying selection due to the ongoing arms race between virus and host immune system. This is visible in the distribution of genotypes at the most repeatedly mutated sites in OC43 lineage A (*Figure 2B and C*).

## Nonsynonymous and synonymous divergence in RdRp and subdomains of spike

An adaptively evolving gene, or region of the genome, should exhibit a high rate of nonsynonymous substitutions. For each seasonal HCoV lineage, we calculated nonsynonymous and synonymous divergence as the average Hamming distance from that lineage's most recent common ancestor (*Zanini et al., 2015*). The rate of nonsynonymous divergence is markedly higher within spike versus RdRp of 229E and OC43 lineage A (*Figure 3A*). While nonsynonymous divergence increases steadily over time in spike, it remains roughly constant at 0.0 in RdRp. These results suggest that there is predominantly positive selection on OC43 and 229E spike, but predominantly purifying selection on RdRp. Separating spike into the S1 (receptor-binding) and S2 (membrane-fusion) domains reveals that the majority of nonsynonymous divergence in spike occurs within S1 (*Figure 3B*). In fact, the

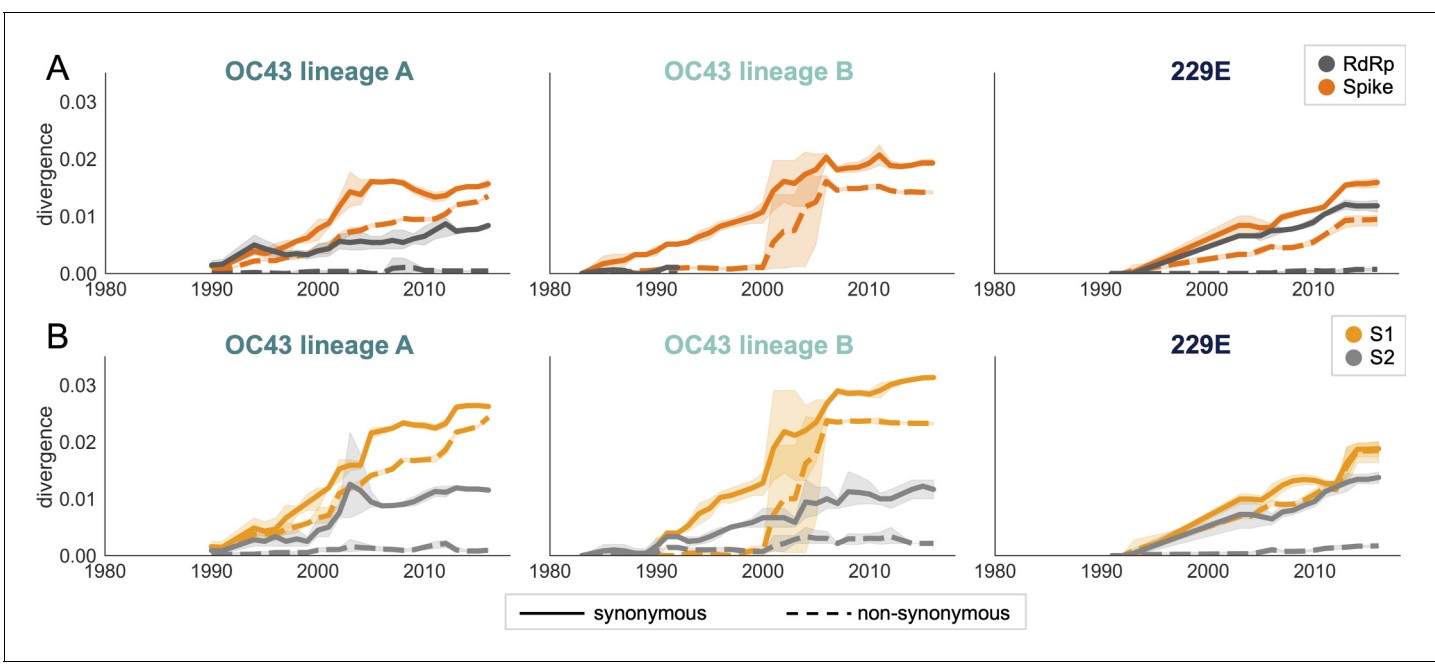

**Figure 3.** Nonsynonymous divergence is higher in OC43 and 229E Spike S1 versus S2 or RdRp. (A) Nonsynonymous (dashed lines) and synonymous divergence (solid lines) of the spike (dark orange) and RdRp (dark gray) genes of all 229E and OC43 lineages over time. Divergence is the average Hamming distance from the ancestral sequence, computed in sliding 3-year windows that contain at least two sequenced isolates. Shaded region shows 95% confidence intervals. Note that the absence of a line means that there are fewer than two sequences available at this time point and that, therefore, the divergence is not calculated. (B) Nonsynonymous and synonymous divergence within the S1 (light orange) and S2 (light gray) domains of spike. Year is shown on the x-axis and is shared between plots.

The online version of this article includes the following figure supplement(s) for figure 3:

**Figure supplement 1.** Nonsynonymous divergence in NL63 and HKU1.

**Figure supplement 2.** Ratio of divergence between genomic regions.

rates of nonsynonymous divergence in S2 are similar to those seen in RdRp, suggesting S2 evolves under purifying selection while S1 evolves adaptively.

Though we would expect synonymous divergence to be equivalent in all areas of the genome, this is not born out in our results. It is unclear whether the difference in synonymous divergence between genes reflects an actual biological difference. However, the ratio of nonsynonymous divergence in spike to nonsynonymous divergence in RdRp is consistently higher than the equivalent ratio of synonymous divergence (*Figure 3—figure supplement 2*). Thus, despite differences in synonymous divergence, spike is accumulating relatively more nonsynonymous divergence than RdRp.

We compared our analysis of divergence to the results of a more standard approach for detecting positive selection on certain branches of a phylogeny. This approach, called mixed effects model of evolution (MEME), is maximum-likelihood method that gives a single *dN/dS* value for each gene (*Murrell et al., 2012*; *Weaver et al., 2018*). In agreement with measures of nonsynonymous divergence over time, *dN/dS* estimates are higher in Spike than RdRp and higher in S1 than S2 (*Table 1*). Our estimate of *dN/dS* in OC43 spike is similar to the previously reported estimate of roughly 0.3 (*Ren et al., 2015*). However, we believe that the standard *dN/dS* approach is not the ideal tool for detecting adaptive evolution in HCoVs because it is a phylogenetic approach, which may be biased by recombination, and also because some assumptions of the model hold true for mammalian genomes, but not necessarily for RNA viruses.

## Rate of adaptation in RdRp and subdomains of spike

Therefore, as a complement to the divergence analysis, we implemented an alternative to the *dN/dS* method that was specifically designed to detect positive selection within RNA virus populations (*Bhatt et al., 2011*). Compared with traditional *dN/dS* methods, the Bhatt method has the advantages of: (1) measuring the strength of positive selection within a population given sequences collected over time, (2) higher sensitivity to identifying mutations that occur only once and sweep through the population, and (3) correcting for deleterious mutations (*Bhatt et al., 2010*; *Bhatt et al., 2011*). Briefly, this method defines a class of neutrally evolving nucleotide sites as those with synonymous mutations or where nonsynonymous polymorphisms occur at medium frequency. Then, the number of fixed and high-frequency nonsynonymous sites that exceed the neutral expectation are calculated. This method compares nucleotide sequences at each time point (the ingroup) to the consensus nucleotide sequence at the first time point (the outgroup) and yields an estimate of the number of adaptive substitutions within a given genomic region at each of these time points.

We adapted this method to detect adaptive substitutions in seasonal HCoVs. As shown in *Figure 4*, OC43 lineage A has continuously amassed adaptive substitutions in spike over the past >30 years while RdRp has accrued few, if any, adaptive substitutions. These adaptive substitutions are located within the S1, and not the S2, domain of spike (*Figure 4*). We observe a largely linear accumulation of adaptive substitutions in spike and S1 through time, although the method does not dictate a linear increase. This observation suggests that spike (and S1 in particular) is evolving in response to a continuous selective pressure. This is exactly what would be expected if these adaptive substitutions are evidence of antigenic evolution resulting from an evolutionary arms race between spike and the host immune system.

We estimate that OC43 lineage A accumulates roughly $1.8 \times 10^{-3}$ adaptive substitutions per codon per year (or 1.4 adaptive amino acid substitutions in S1 each year) in the S1 domain of spike, while the rate of adaptation in OC43 lineage B is slightly higher and is estimated to result in an

---

**Table 1.** *dN/dS* is lower in spike than RdRp.
A single *dN/dS* value was computed for gene (or spike domain) and each human coronavirus (HCoV) using mixed effects model of evolution (MEME).

|                | RdRp  | Spike | S1    | S2    |
|----------------|-------|-------|-------|-------|
| 229E           | 0.143 | 0.441 | 0.662 | 0.166 |
| OC43 lineage A | 0.080 | 0.435 | 0.466 | 0.301 |
| OC43 lineage B | 0.061 | 0.317 | 0.418 | 0.234 |
| NL63           | 0.068 | 0.139 | 0.121 | 0.038 |

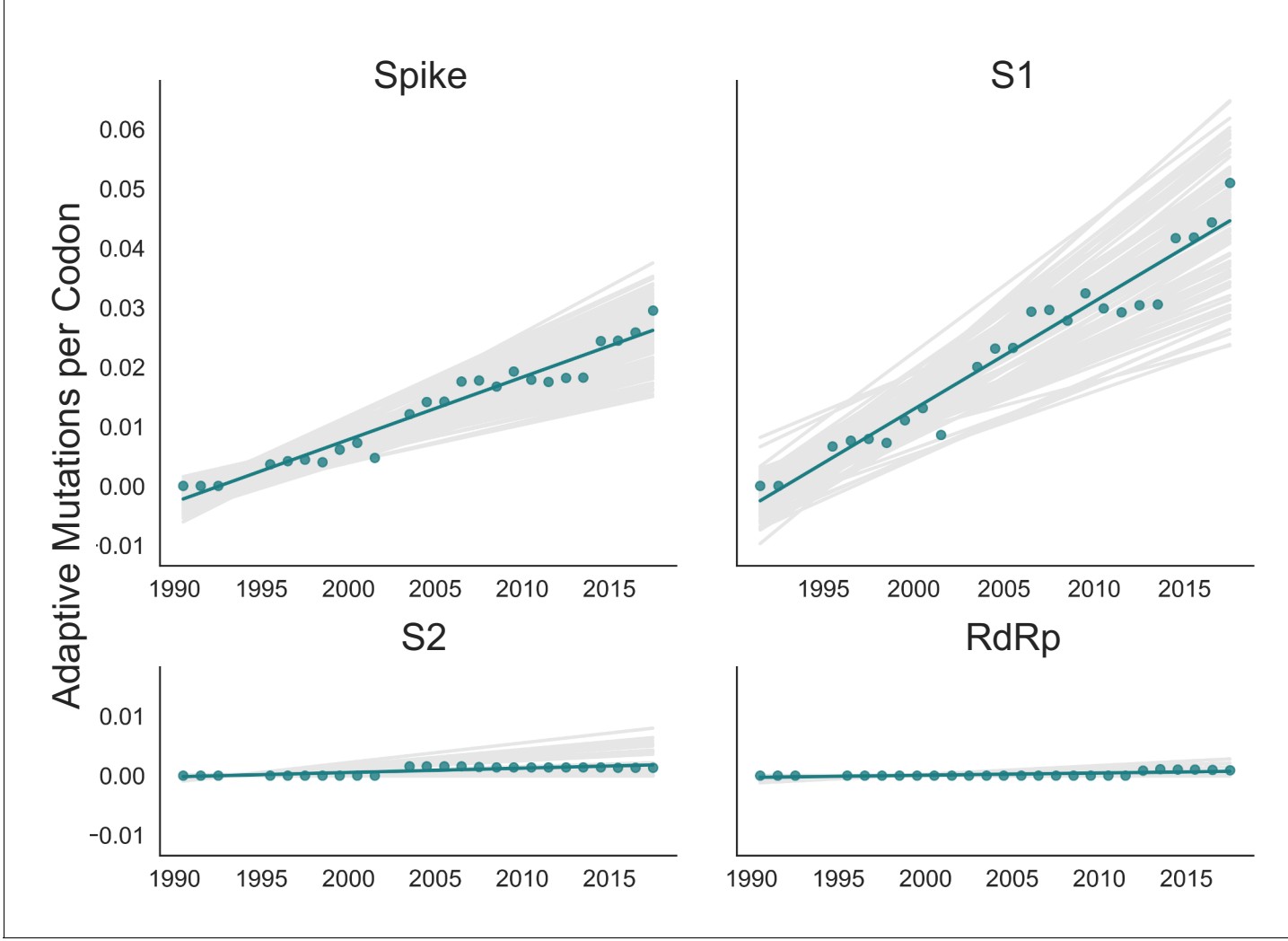

**Figure 4.** Adaptive substitutions accumulate over time in OC43 lineage A spike S1. Adaptive substitutions per codon within OC43 lineage A spike, S1, S2 and RdRp as calculated by our implementation of the Bhatt method. Adaptive substitutions are computed in sliding 3-year windows, and only for time points that contain three or more sequenced isolates. Green dots display estimated values calculated from the empirical data and green lines show linear regression fit to these points. Gray lines show the distribution of regressions fit to the computed number of adaptive substitutions from 100 bootstrapped data sets. Year is shown on the x-axis.

average 1.7 adaptive substitutions in S1 per year (*Figure 5*). The S1 domain of 229E is estimated to accrue 0.76 adaptive substitutions per year (a rate of $1.4 \times 10^{-3}$ adaptive substitutions per codon per year).

A benefit of the Bhatt method is the ability to calculate the strength of selection, which allows us to compare these seasonal HCoVs to other viruses. We used our implementation of the Bhatt method to calculate the rate of adaptation for influenza A/H3N2, which is known to undergo rapid antigenic evolution (*Rambaut et al., 2008*; *Yang, 2000*), measles, which does not (*Fulton et al., 2015*), and influenza B strains Vic and Yam, which evolve antigenically at a slower rate than A/H3N2 (*Bedford et al., 2014*). We estimate that the receptor-binding domain of influenza A/H3N2 accumulates adaptive substitutions between two and three times faster than the HCoVs OC43 and 229E (*Figure 6*). The rates of adaptive substitution in influenza B/Yam and B/Vic are on par with the seasonal HCoVs. We detect no adaptive substitutions in the measles receptor-binding protein. These results put the evolution of the S1 domain of OC43 and 229E in context, indicating that the S1 domain is under positive selection, and that this positive selection generates new variants in the putative antigenic regions of these HCoVs at about the same rate as influenza B strains and about half the rate of the canonical example of antigenic evolution, the HA1 domain of influenza A/H3N2.

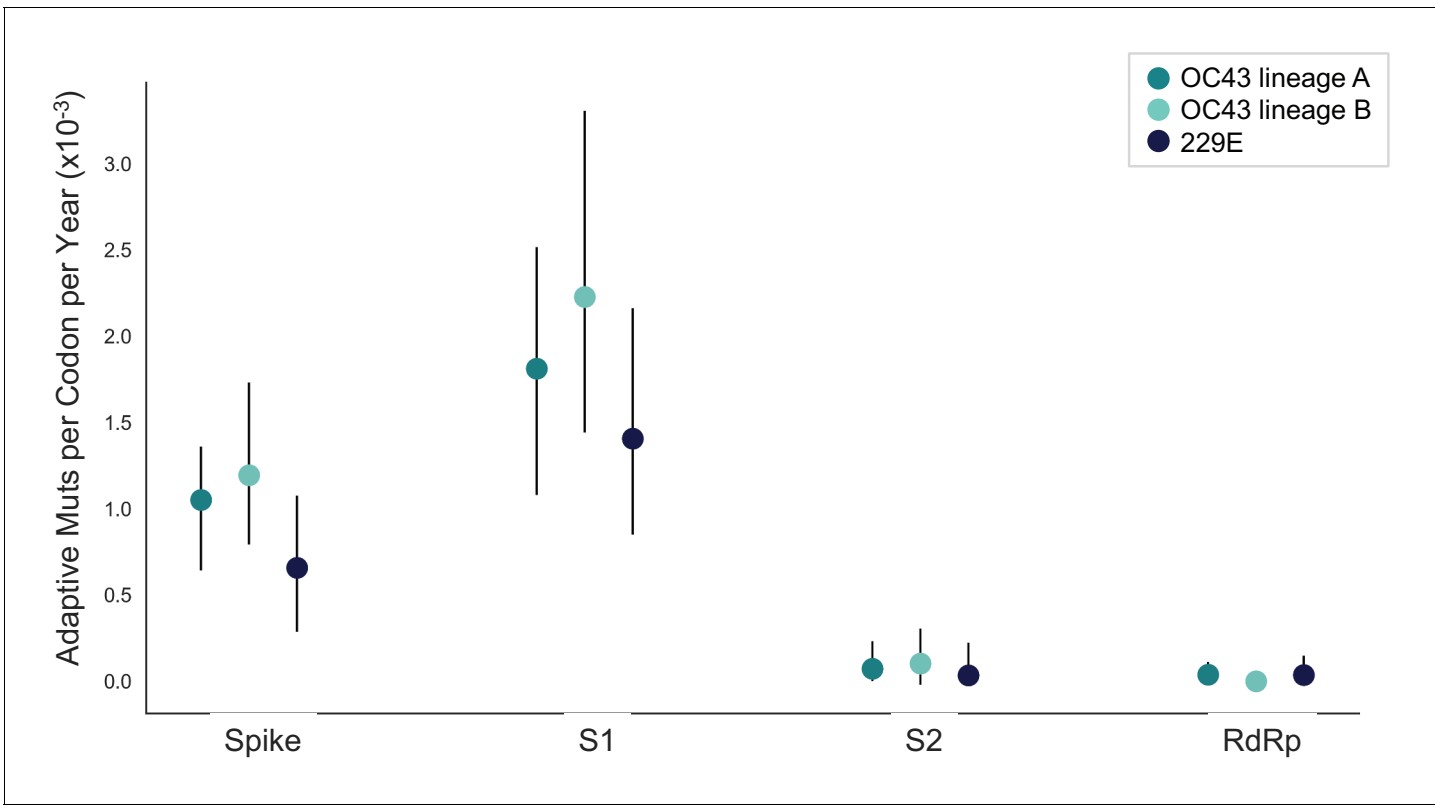

**Figure 5.** The rate of adaptive substitution is highest in spike S1. Adaptive substitutions per codon per year as calculated by our implementation of the Bhatt method. Rates are calculated within Spike, S1, S2 and RdRp for 229E and OC43 lineages. Error bars show 95% bootstrap percentiles from 100 bootstrapped data sets.

The online version of this article includes the following figure supplement(s) for figure 5:

**Figure supplement 1.** NL63 and HKU1 have low rates of adaptation in spike.

## Validation that rate of adaptation is not biased by recombination

Because coronaviruses are known to recombine, and recombination has the potential to impact evolutionary analyses of selection, we sought to verify that our results are not swayed by the presence of recombination. To do this, we simulated the evolution of OC43 lineage A spike and RdRp genes under varying levels of recombination and positive selection (representative phylogenies of simulated spike evolution can be seen in *Figure 7—figure supplement 2*) and used our implementation of the Bhatt method to identify adaptive evolution. As the strength of positive selection increases, we detect a higher rate of adaptive evolution, regardless of the level of recombination (*Figure 7*). This demonstrates that our estimates of adaptive evolution are not biased by recombination events.

## TMRCA of RdRp and subdomains of spike

Finally, we know that strong directional selection skews the shape of phylogenies (*Volz et al., 2013*). In influenza H3N2, immune selection causes the genealogy to adopt a ladder-like shape where the rungs are formed by viral diversification and each step is created by the appearance of new, antigenically superior variants that replace previous variants. This ladder-like shape can also be seen in the phylogenies of the OC43 and 229E (*Figure 1*). In this case, selection can be quantified by the timescale of population turnover as measured by the TMRCA, with the expectation that stronger selection will result in more frequent steps and therefore a smaller TMRCA measure (*Bedford et al., 2011*). We computed average TMRCA values from phylogenies built on Spike, S1, S2 or RdRp sequences of OC43 lineage A and 229E (*Table 2*). We did not compute TMRCA for OC43 lineage B because the limited number of available RdRp sequences for this lineage mean that TMRCA can only be calculated for about 4 years, which could artificially skew the TMRCA estimates. Our

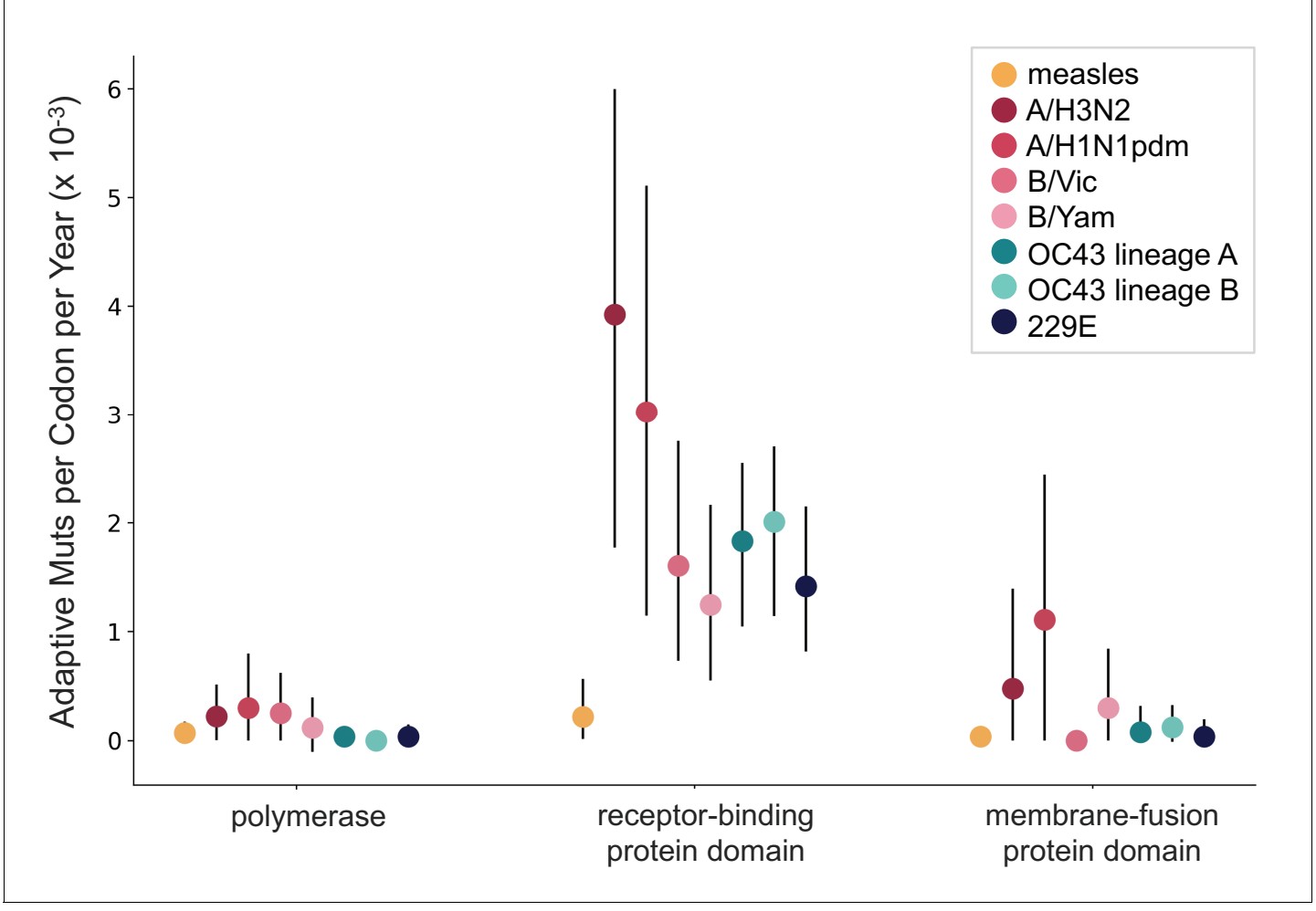

**Figure 6.** OC43 and 229E spike S1 accumulates adaptive substitutions faster than measles but slower than influenza A/H3N2. Comparison of adaptive substitutions per codon per year between measles (yellow), four influenza strains (A/H3N2, A/H1N1pdm, B/Vic, and B/Yam- shown in shades of red), OC43 lineage A (dark teal), OC43 lineage B (light teal), and 229E (dark blue). The polymerase, receptor-binding domain, and membrane fusion domain for influenza strains are PB1, HA1, and HA2. For both human coronaviruses (HCoVs), they are RdRp, S1, and S2, respectively. For measles, the polymerase is the L gene, the receptor-binding protein is the H gene, and the fusion protein is the F gene. Error bars show 95% bootstrap percentiles from 100 bootstrapped data sets.

estimates of HCoV spike TMRCA are roughly 2–2.5 longer the estimated TMRCA for influenza A/H3N2 hemagglutinin (*Bedford et al., 2011*).

We observe that for both OC43 lineage A and 229E, the average TMRCA is lower in spike than RdRp and lower in S1 versus S2. These results suggest strong directional selection in S1, likely driven by pressures to evade the humoral immune system. The difference in TMRCA between S1 and S2 is indicative not only of differing selective pressures acting on these two spike domains, but also of intra-spike recombination. This is because the immune selection imposed on S1 should also propagate neutral hitch-hiker mutations in closely linked regions such as S2. The difference in TMRCA suggests that recombination may uncouple these regions. Recombination can also push TMRCA to higher values, though this should not have a larger impact on RdRp than S1. The contributions of the forces of directional selection and recombination are difficult to parse from the TMRCA results. This emphasizes the importance of using methods, such as the Bhatt method, that are robust to recombination to detect adaptive evolution.

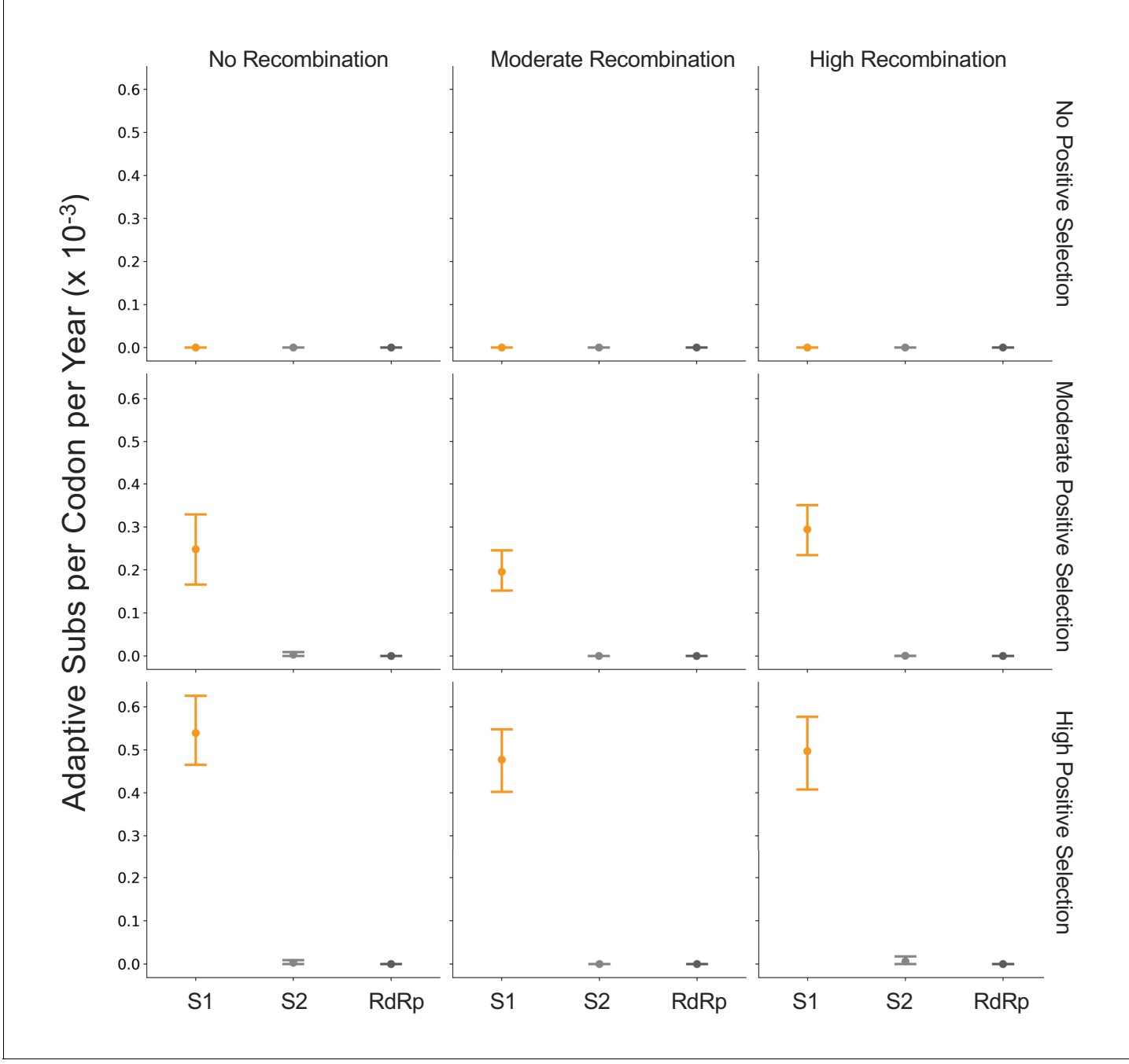

**Figure 7.** Detection of positive selection is not biased by recombination. OC43 lineage A sequences were simulated with varying levels of recombination and positive selection. The Bhatt method was used to calculate the rate of adaptive substitutions per codon per year for S1 (light orange), S2 (light gray), and RdRp (dark gray). The mean and 95% confidence interval of 10 independent simulations are plotted.

The online version of this article includes the following figure supplement(s) for figure 7:

**Figure supplement 1.** Fewer years of longitudinally sampled isolates reduces ability to detect rate of adaptation.

**Figure supplement 2.** Representative phylogenies of simulated spike data.

## Application of methods for identifying adaptive evolution to HKU1 and NL63

Because HKU1 was identified in the early 2000s, there are fewer longitudinally sequenced isolates available for this HCoV compared to 229E and OC43 (*Figure 1—figure supplement 2*).

**Table 2.** Mean Time to Most Recent Common Ancestor (TMRCA) is lower in S1 than RdRp or S2. Average TMRCA values (in years) for OC43 lineage A and 229E. The 95% confidence intervals are indicated in parentheses below mean TMRCA values.

|  | Spike | S1 | S2 | RdRp |
|---|---|---|---|---|
| OC43 lineage A | 4.67 (4.04, 5.28) | 3.45 (2.86, 4.05) | 13.05 (11.24, 14.97) | 17.39 (15.63, 19.15) |
| 229E | 4.19 (3.13, 5.25) | 2.23 (1.76, 2.69) | 5.08 (3.93, 6.23) | 4.86 (4.04, 5.69) |

Consequently, the phylogenetic reconstructions and divergence analysis of HKU1 have a higher level of uncertainty. To begin with, it is less clear from the phylogenies whether HKU1 represents a single HCoV lineage like 229E or, instead, should be split into multiple lineages like OC43 (*Figure 1*). Because of this, we completed all antigenic analyses for HKU1 twice: once considering all isolates to be members of a single lineage, and again after splitting isolates into two separate lineages. These lineages are arbitrarily labeled 'A' and 'B' as was done for OC43. When HKU1 is considered to consist of just one lineage, there is no signal of antigenic evolution by divergence analysis (*Figure 3—figure supplement 1B*) or by the Bhatt method of estimating adaptive evolution (*Figure 5—figure supplement 1A*). However, when HKU1 is assumed to consist of two co-circulating lineages, HKU1 lineage A has a markedly higher rate of adaptive substitutions in S1 than in S2 or RdRp (*Figure 5—figure supplement 1B*).

To demonstrate the importance of having a well-sampled longitudinal series of sequenced isolates for our antigenic analyses, we returned to our simulated OC43 S1 data sets. We mimicked shorter longitudinal series by truncating the data set to only 24, 14, 10, or 7 years of samples and ran the Bhatt analysis on these sequentially shorter time series (*Figure 7—figure supplement 1*). The rates of adaptation estimated from the truncated data sets can be compared to the 'true' rate of adaptation calculated from all simulated data. This simulated data reveals a general trend that less longitudinal data reduces the ability to detect adaptive evolution by skewing the estimated rate away from the 'truth' and increasing the uncertainty of the analysis. Given the dearth of longitudinal data for HKU1, we do not feel that it is appropriate to make strong conclusions about adaptive evolution, or lack thereof, in this HCoV.

Despite being identified at roughly the same time as HKU1, substantially more NL63 isolates have been sequenced (*Figure 1—figure supplement 2*) making the phylogenetic reconstruction and evolutionary analyses of this virus correspondingly more reliable. We do not observe evidence for adaptive evolution in NL63 (*Figure 3—figure supplement 1A* and *Figure 5—figure supplement 1A*) and this lack of support for adaptive evolution in the NL63 spike gene is more likely to reflect an actual lack of adaptive evolution in this virus.

## Discussion

Using several corroborating methods, we provide evidence that the seasonal HCoVs OC43 and 229E undergo adaptive evolution in S1, the region of the spike protein exposed to human humoral immunity (*Figures 3*, *4*, and *5*). We additionally confirm that RdRp and S2 do not show signals of adaptive evolution. We observe that S1 accumulates between 0.8 (229E) and 1.4 (OC43) adaptive substitutions per year. We infer that these viruses accumulate adaptive substitutions at roughly half the rate of influenza A/H3N2 and at a similar rate to influenza B viruses (*Figure 6*). The most parsimonious explanation for the observation of substantial adaptive evolution in S1 is that antigenic drift is occurring in which mutations that escape from human population immunity are selectively favored in the viral population leading to repeated adaptive changes. However, it is formally possible that the adaptive evolution we detect is a result of selective pressures other than evasion of the adaptive immune system. Showing that this is truly antigenic evolution could involve a serological comparison of isolates that differ at S1 residues under positive selection.

In seasonal influenza and measles, the rates of adaptive evolution we estimate correlate well with relative rates of antigenic drift reported by other groups (*Fulton et al., 2015*; *Bedford et al., 2014*). The relative rates of adaptation we calculate also match the relative frequency of vaccine strain

updates, as would be expected since vaccines must be updated to match antigenically evolving viruses. Since 2006, the A/H3N2 component of the seasonal influenza vaccine has been updated 10 times (11 different A/H3N2 strains), four different B/Vic strains and four different B/Yam strains have been included in the vaccine, and the measles vaccine strain has not changed (Global Influenza Surveillance and Response System [GISRS], https://www.who.int/influenza/vaccines/virus/en/). Using these numbers as guidance, our results suggest that a vaccine against OC43 or 229E might need to be updated as frequently as the B/Vic and B/Yam components of the influenza vaccine are.

We do not observe evidence of antigenic evolution in NL63 (*Figure 3—figure supplement 1* and *Figure 5—figure supplement 1*). This likely represents a lack of marked adaptive evolution in S1. Our finding fits with a study of NL63 in Kenya, which identified multiple genotypes of NL63 and show that people regularly become reinfected with the same genotype of NL63 (*Kiyuka et al., 2018*). Additionally, Kiyuka et al. found that these genotypes circulate locally for a long period of time, suggesting a decent amount of viral diversity and a potential lack of evolution due to immune selection. Though our results cannot explain why OC43 and 229E likely evolve antigenically while NL63 does not, Kiyuka et al. observe that NL63 reinfections are sometimes enhanced by a previous infection and hypothesize that NL63 is actually under purifying selection at epitope sites (*Kiyuka et al., 2018*).

Though analysis of all HCoVs would benefit from more sequenced isolates, there is substantially less longitudinal sequencing data available for HKU1. Thus, despite finding no evidence of antigenic evolution in HKU1 (*Figure 3—figure supplement 1* and *Figure 5—figure supplement 1*), it is possible that a more completely sampled time series of HKU1 genome sequences could alter the result for this virus (*Figure 7—figure supplement 1*).

Our conclusions of adaptive evolution in S1, arrived at through computational analyses of sequencing data, agree with studies that observe reinfection of subjects by heterologous isolates of 229E (*Reed, 1984*), sequential dominance of specific genotypes of OC43 (*Lau et al., 2011*; *Zhang et al., 2015*), and common reinfection by seasonal HCoVs from longitudinal serological data (*Edridge et al., 2020*). In this latter study, HCoV infections were identified from longitudinal serum samples by assaying for increases in antibodies against the nucleocapsid (N) protein of representative OC43, 229E, HKU1, and NL63 viruses. This study concluded that the average time between infections was 1.5–2.5 years, depending on the HCoV (*Edridge et al., 2020*). In comparison, influenza H3N2 reinfects people roughly every 5 years (*Kucharski et al., 2015*). Thus, frequent reinfection by seasonal HCoVs is likely due to a combination of factors and suggests waning immune memory, and/or incomplete immunity against reinfection, in addition to antigenic drift.

HCoVs are a diverse grouping split, phylogenetically, into two genera: NL63 and 229E are alphacoronaviruses, while OC43, HKU1, MERS, SARS, and SARS-CoV-2 are betacoronaviruses. The method of cell entry does not seem to correlate with genus. Coronaviruses bind to a remarkable range of host-cell receptors including peptidases, cell adhesion molecules, and sugars. Among the seasonal HCoVs, OC43 and HKU1 both bind 9-O-acetylsialic acid (*Hulswit et al., 2019*), while 229E binds human aminopeptidase N (hAPN) and NL63 binds angiotensin-converting enzyme 2 (ACE2) (*Liu et al., 2020b*). Despite a relatively large phylogenetic distance and divergent S1 structures, NL63 and SARS-CoV-1 and SARS-CoV-2 bind to the same host receptor using the same virus-binding motifs (VBMs) (*Li, 2016*). This VBM is located in the C-terminal domain of S1 (S1-CTD), which fits within the trend of S1-CTD receptor-binding in CoVs that bind protein receptors (*Hofmann et al., 2006*; *Li, 2016*). This is opposed to the trend among CoVs that bind sugar receptors, where receptor binding is located within the S1-NTD (*Li, 2016*). This localization roughly aligns with our observations that the majority of the repeatedly mutated sites occur toward the C-terminal end of 229E S1 and the N-terminal end of OC43 S1 (*Figure 2*).

Here we have provided support that at least two of the four seasonal HCoVs evolve adaptively in the region of spike that is known to interact with the humoral immune system. These two viruses span both genera of HCoVs, though due to the complexity of HCoV receptor binding and pathology mentioned above, it is not clear whether or not this suggests that other HCoVs, such as SARS-CoV-2, will also evolve adaptively in S1. This is important because, at the time of writing of this manuscript, many SARS-CoV-2 vaccines are in production and most of these exclusively include spike (*Krammer, 2020*). If SARS-CoV-2 evolves adaptively in S1 as the closely related HCoV OC43 does, it is possible that the SARS-CoV-2 vaccine would need to be frequently reformulated to match the circulating strains, as is done for seasonal influenza vaccines.

## Materials and methods

All data, source code, and analyses can be found at https://github.com/blab/seasonal-cov-adaptive-evolution (*Kistler, 2021*; copy archived at swh:1:rev: 83721cd000f2848d4f77d2a6da8c2d0df8a555a1). All phylogenetic trees constructed and analyzed in this manuscript can be viewed interactively at https://nextstrain.org/community/blab/seasonal-cov-adaptive-evolution; *Mattenberger, 2021*. All analysis code is written in Python 3 (Python Programming Language, SCR_008394) in Jupyter notebooks (Jupyter-console, RRID:SRC_018414).

### Sequence data

All viral sequences are publicly accessible and were downloaded from ViPR (http://www.viprbrc.org) under the 'Coronavirdiae' with host 'human' (*Pickett et al., 2012*). Sequences labeled as 'OC43', '229E', 'HKU1', and 'NL63' were pulled out of the downloaded FASTA file into four separate data files. Additionally, a phylogeny of all downloaded HCoVs was made and unlabeled isolates that clustered within clades formed by labeled OC43, 229E, HKU1, or NL63 isolates were marked as belonging to that HCoV type and added to our data files. Code for these data-parsing steps is located in `data-wrangling/postdownload_formatting_for_rerun.ipynb`.

### Phylogenetic inference

For each of the four HCoV data sets, full-length sequences were aligned to a reference genome using the augur align command (*Hadfield et al., 2018*) and MAFFT (*Katoh et al., 2002*). Individual gene sequences were then extracted from these alignments if sequencing covered 50% or more of the gene using the code in `data-wrangling/postdownload_formatting_for_rerun.ipynb`. Sequence files for each gene are located in the data/ directory within each HCoV parent directory (ex: `oc43/data/oc43_spike.fasta`). A Snakemake file (*Köster and Rahmann, 2012*) within each HCoV directory follows the general outline of a Nextstrain build (Nextstrain, RRID:SCR_018223) and was used to align each gene to a reference strain and build a time-resolved phylogeny with IQ-Tree v1 (*Nguyen et al., 2015*) and TimeTree (*Sagulenko et al., 2018*). Phylogenies were viewed to identify the distribution of genotypes throughout the tree, different lineages, and signals of recombination using the nextstrain view command (*Hadfield et al., 2018*). The clock rate of the phylogeny based on spike sequences for each isolate (as shown in *Figure 1* and *Figure 1—figure supplement 2*) was 0.0005 substitutions per nucleotide site per year for OC43, 0.0006 for 229E, 0.0007 for NL63, and 0.0062 for HKU1. All NL63 and HKU1 trees were rooted on an outgroup sequence. For NL63, the outgroup was 229e/AF304460/229e_ref/Germany/2000 and for HKU1 the outgroup was mhv/NC_048217_1/mhv/2006. Clock rates for the phylogenies built on each individual gene can be found within the `results/` directory within each HCoV parent directory (ex: `oc43/results/branch_lengths_oc43_spike.json`).

### Mutation counting

Amino acid substitutions at each position in spike were tallied from the phylogeny. In other words, the phylogenetic reconstruction of spike sequences returns nucleotides changes to the ancestral sequence along each branch. The number of times this changed amino acid identity at each position was tallied. This analysis was conducted using code in `antigenic_evolution/site_mutation_rank.ipynb`.

### Divergence analysis

For each HCoV lineage and each gene, synonymous and nonsynonymous divergence was calculated at all time points as the average Hamming distance between each sequenced isolate and the consensus sequence at the first time point (founder sequence). The total number of observed differences between the isolate and founder nucleotide sequences that result in nonsynonymous (or synonymous) substitutions is divided by the number of possible nucleotide mutations that result in nonsynonymous (or synonymous) substitutions, weighted by kappa, to yield an estimate of divergence. Kappa is the ratio of rates of transitions:transversions and was calculated by averaging values from spike and RdRp trees built by BEAST 2.6.3 (*Bouckaert et al., 2019*) using the HKY+gamma4 model with two partitions and 'coalescent constant population'. All BEAST results are found in. `log` files in gene- and HCoV-specific subdirectories within `beast/`. Divergence is calculated from

nucleotide alignments. Sliding 3-year windows were used and only time points that contained at least two sequences were considered. The concept for this analysis is from *Zanini et al., 2015* and code for our adaptation is in `antigenic_evolution/divergence_weighted.ipynb`. The ratios of divergence shown in *Figure 3—figure supplement 2* are also calculated in this notebook.

## Calculation of *dN/dS*

A *dN/dS* value was calculated for RdRp, spike, S1 and S2 of each HCoV using the Datamonkey (*Weaver et al., 2018*) implementation of MEME (*Murrell et al., 2012*). Aligned FASTA files (ex: `oc43/results/aligned_oc43_rdrp.fasta`) were uploaded to Datamonkey (http://datamonkey.org/meme) and *dN/dS* value was recorded as the calculated Global MG94xREV model non-synonymous/synonymous rate ratio.

## Implementation of the Bhatt method

The rate of adaptive evolution was computed using an adaptation of the Bhatt method (*Bhatt et al., 2011*; *Bhatt et al., 2010*). For each lineage and each genomic region, we partitioned all available sequences into sliding 3-year windows and only used time points that contained at least three sequences in the analysis. We compared nucleotide sequences at each time point (the ingroup) to the consensus nucleotide sequence at the first time point (the outgroup). Eight estimators (silent fixed, replacement fixed, silent high frequency, replacement high frequency, silent mid-frequency, replacement mid-frequency, silent low frequency, and replacement low-frequency) are calculated by the site-counting method (*Bhatt et al., 2010*). In the site-counting method, each estimator is the product of the fixation or polymorphism score times the silent or replacement score, summed for each site in that frequency class. Fixation and polymorphism scores depend on the number of different nucleotides observed at the site and whether the outgroup base is present in the ingroup. Selectively neutral sites are assumed to contain the classes of silent polymorphisms and replacement polymorphisms occurring at a frequency between 0.15 and 0.75. A class of nonneutral, adaptive sites is then identified as having an excess of replacement fixations or polymorphisms (*Bhatt et al., 2011*). For each lineage and gene, 100 bootstrap alignments and ancestral sequences were generated and run through the Bhatt method to assess the statistical uncertainty of our estimates of rates of adaptation (*Bhatt et al., 2011*). The rate of adaptation (per codon per year) shown in *Figure 5* is calculated by linear regression of the time series values of adaptive substitutions per codon (*Figure 4*). Our code for implementing the Bhatt method is at `antigenic_evolution/bhatt_bootstrapping.ipynb`.

## Estimation of rates of adaptation of measles and influenza viruses

Influenza and measles alignments were generated by running Nextstrain the respective Nextstrain builds from https://github.com/nextstrain/seasonal-flu and https://github.com/nextstrain/measles (*Hadfield et al., 2018*; Nextstrain, RRID:SCR_018223). The seasonal influenza build was run with 20-year resolution for H3N2, H1N1pdm, Vic, and Yam. The rates of adaptation of different genes was calculated using our implementation of the Bhatt method described above. The receptor-binding domain used for influenza was HA1, for measles was the H protein, and for the HCoVs was S1. The membrane fusion protein used for influenza was HA2, for measles was the F protein, and for the HCoVs was S2. The polymerase for influenza was PB1, for measles was the L protein, and for the HCoVs was RbRd (nsp12). Our code for this analysis is at `antigenic_evolution/bhatt_nextstrain.ipynb`.

## Simulation of evolving OC43 sequences

The evolution of OC43 lineage A Spike and RdRp genes was simulated using SANTA-SIM (*Jariani et al., 2019*). The OC43 lineage A root sequence was used as a starting point and the simulation was run for 500 generations and 10 simulated sequences were sampled every 50 generations. The spike and RdRp genes were simulated separately. Purifying selection was simulated across both genes. Evolution was simulated in the absence of recombination and with moderate and high levels of recombination during replication. Under each of these recombination paradigms, we simulated evolution in the absence of positive selection within spike and with moderate and high levels of positive selection. Positive selection was simulated through exposure-dependent selection at a subset of

spike S1 sites proportional to the number of epitope sites in H3N2 HA (*Luksza and Lässig, 2014*). The simulated selection allows mutations in these 'epitope' sites to rise in frequency while also encouraging 'epitopes' to change over time (to mimic antigenic novelty). All simulations were run with a nucleotide mutation rate of $1 \times 10^{-4}$ (*Vijgen et al., 2005*). Config files, results, and source code for these simulations can be at `santa-sim_oc43a/` and the Bhatt method is implemented on the simulated data in `antigenic_evolution/bhatt_simulated_oc43_data.ipynb`.

### Estimation of TMRCA

Mean TMRCA values were estimated for each gene and each HCoV using PACT (*Bedford et al., 2011*). Briefly, PACT computes TMRCA values by creating a series of subtrees that include only tips positioned within a temporal slice of the full tree and finding the common ancestor of these tips. The overall mean and 95% confidence intervals were calculated from the list of TMRCA values in these time slices. The PACT config files and results for each run are in the directory `antigenic_evolution/pact/`. The TMRCA estimations and subsequent analyses are executed by code in `antigenic_evolution/tmrca_pact.ipynb`.

## Acknowledgements

We thank Jesse Bloom and members of the Bedford lab for useful feedback. KEK was supported by the National Science Foundation Graduate Research Fellowship Program under Grant No. DGE-1762114. TB is a Pew Biomedical Scholar and is supported by NIH R35 GM119774-01.

## Additional information

### Funding

| Funder | Grant reference number | Author |
|---|---|---|
| National Science Foundation | Graduation Research Fellowship Program | Kathryn E Kistler |
| Pew Charitable Trusts | Pew Biomedical Scholar | Trevor Bedford |
| National Science Foundation | DGE-1762114 | Kathryn E Kistler |
| Pew Charitable Trusts | NIH R35 GM119774-01 | Trevor Bedford |

The funders had no role in study design, data collection and interpretation, or the decision to submit the work for publication.

### Author contributions

Kathryn E Kistler, Conceptualization, Data curation, Software, Formal analysis, Funding acquisition, Validation, Investigation, Visualization, Methodology, Writing - original draft, Writing - review and editing; Trevor Bedford, Conceptualization, Supervision, Funding acquisition, Validation, Methodology, Writing - review and editing

### Author ORCIDs

Kathryn E Kistler https://orcid.org/0000-0002-3216-0020
Trevor Bedford http://orcid.org/0000-0002-4039-5794

### Decision letter and Author response

Decision letter https://doi.org/10.7554/eLife.64509.sa1
Author response https://doi.org/10.7554/eLife.64509.sa2

## Additional files

### Supplementary files

• Transparent reporting form

## Data availability

All data used in this study can be found at https://www.viprbrc.org/ and is archived as .fasta files in the following subdirectories of the Github repository for this project: https://github.com/blab/seasonal-cov-adaptive-evolution/tree/master/229e/data, https://github.com/blab/seasonal-cov-adaptive-evolution/tree/master/oc43/data, https://github.com/blab/seasonal-cov-adaptive-evolution/tree/master/nl63/data, https://github.com/blab/seasonal-cov-adaptive-evolution/tree/master/hku1/data (copy archived at https://archive.softwareheritage.org/swh:1:rev:83721cd000f2848d4f77d2a6da8c2d0df8a555a1).

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
