## [Decision Letter]

**Acceptance summary:**

Coronavirus adaptation to immune responses is a very timely and important topic. This paper demonstrates that some of the seasonal coronavirus strains responsible for common colds have undergone substantial recent adaptation in a protein that is likely targeted by the immune system. However, they have not been adapting as quickly as similar proteins in the influenza virus, and at least one seasonal coronavirus strain has undergone very little adaptation in this protein.

**Decision letter after peer review:**

Congratulations, we are pleased to inform you that your article, "Evidence for adaptive evolution in the receptor-binding domain of seasonal coronaviruses", has been accepted for publication in eLife.

Please take note of the points below and we hope you will continue to support eLife. The main point that all the reviewers agreed on is that it would be helpful to compare your results to those that would be obtained with a standard dN/dS approach.

*Reviewer #1:*

The authors use dN/dS-based statistics to show that the S1 domain has been adapting in coronaviruses OC43 and 229E, but not in NL63. For at least OC43 and 229E, this agrees qualitatively with earlier studies. The authors should make a detailed, quantitative comparison with these earlier results.

1) I had a hard time understanding exactly what was going on in the simulations, and whether, e.g., they produced phylogenies that looked like the real ones. But perhaps the simulations should be cut or de-emphasized. From my understanding of the Bhatt et al. papers and the authors' code, their selection-detection method is entirely site-based and doesn't use the tree at all, and so should be insensitive to recombination a priori. If this is correct, the simulations are largely unnecessary (but see the next point).

2) It looks like there is substantial geographic population structure, as well as time-varying geographic biases in sampling. This seems like it could generate false signals of substitutions, where standing spatial variation appears to be temporal variation. Can the authors bound how large this contribution could be? If not, perhaps they could still say that they've found signatures of either global positive selection or local adaptation, both of which are interesting.

3) Why is synonymous divergence also much higher in OC43 S1 than it is in RDRP or S2?

4) As I mention in the general assessment above, the authors should compare their results to previous studies. This includes both overall substitution rates as well as specific sites found to have repeated substitutions. For NL63, how does the null result compare to what's known? I believe Kiyuka et al., 2018, found widespread reinfection by similar NL63 genotypes, which may be relevant.

*Reviewer #2:*

The paper by Kistler and Bedford explores whether adaptive evolution has led to diversification of coronaviruses responsible for the common cold in the human population. The paper is very well written and presents evidence for adaptive evolution in some strains and lack of evidence in others. I enjoyed reading the paper.

*Reviewer #3:*

This paper analyzing potential adaptive evolution in seasonal coronaviruses ("common colds") is highly relevant with well-supported conclusions. Its results will be widely applicable to myriad fields, including evolutionary biology, epidemiology and public health, vaccinology, immunology, and virology.

Kistler and Bedford present a timely and highly relevant analysis of adaptive evolution in seasonal "common cold" coronaviruses. Overall, I find the research compelling, well-performed, and mostly well-presented. The research contains sufficient statistical rigor with commendable computational reproducibility to be considered highly reliable. The authors conclude that at least two of the four known common cold coronaviruses have been undergoing adaptive evolution in their human hosts. These results may shed light on the emerging long-term evolutionary dynamics of SARS-CoV-2, the causative agent of COVID-19, as it continues circulating in humans, as well as informing ongoing vaccine design.

Kistler and Bedford present a timely and highly relevant analysis of adaptive evolution in seasonal "common cold" coronaviruses. Overall, I find the research compelling, well-performed, and mostly well-presented (though some organizational changes are needed). As always, the commitment to open code and data from the Bedford lab is admirable and successfully performed/communicated. In my comments below, I offer advice to improve the clarity and presentation of the paper, with a few small requested analyses or need for further explanations.

1) "We have arbitrarily labeled these lineages 'A' and 'B' (Figure 1)." The figure then shows A/B panels for different hCoV strains, which is a little confusing at first until you orient to the figure presentation. I recommend labeling the phylogeny in panel A with "A" and "B" rather than just using colors, and including that lineage information in the legend.

2) The legend for Figure 1 needs units for the clock rate. Presumably this is the codon sub rate per year, which is used elsewhere? Or is this nucleotide? Similarly, units are also needed:…

– Subsection “Rate of Adaptation in RdRp and subdomains of spike”, for the parenthetical "(or 0.45 adaptive substitutions each year)". Are the authors converting to nucleotide?

– Subsection “Phylogenetic Inference”.

3) In general, I find the references to supplementary figures in the text confusing. For example, I first read the phrase "Figure 1 Supplement 1A" to mean both Figure 1 and Supplement 1A. Writing this as, "Figure 1—figure supplement 1A" will make it more clear that Figure 1 of the main text is not being referenced.

4) Results, second paragraph – sentences are not well ordered. Should be in order: 1) Though…, 2) Because…, 3) This….

5) The last two sentences in the second paragraph of the Results seem tacked on and not immediately relevant to recombination. Please move these sentences or include a paragraph break.

6) Related to the previous comment, the Results section as a whole will benefit from improved organization, specifically by creating subsections. I highly recommend adding these to improve readability.

7) Comments for Figure 2:

– Unless it becomes too busy, small indicators (or at least in the legend to avoid figure noise) might be added to emphasize the RBD domain within S1 specifically.

– I recommend changing the color of the asterisks in panel A to match the lineage A (red) color.

8) The authors write, "from that lineage's common ancestor." It would be more precise to say the "from that lineage's *most recent* common ancestor."

9) Figure 3, specifically in panel A for RdRp, leaves some ambiguous interpretations: Is the line missing for OC43 lineage B because there is no RdRp data after the early 1990s (seems unlikely?), or because there were no adaptive substitutions, in which case the orange lines should remain steady at 0? This aspect of the figure should be clarified or fixed.

There is a similar situation for panel C, HKU1 lineage B, in Figure 2—figure supplement 1. In addition, the "C" for that panel is cut off at the bottom, so this figure needs to be slightly reformatted.

10) In the fifth paragraph of the Results, the authors introduce the H3N2 analysis, but the actual analysis is not really presented or described for another two pages. The H3N2 comparison is definitely not part of Figure 4, which this paragraph is introducing. I think this sentence is likely misplaced? Again, this comment shows that adding subsections to the Results section will be helpful for overall organization.

11) Results, I would like to see more details about what constitutes an “adaptive substitution” in the Bhatt method, which is not as widely used as dN/dS, within the main text itself. A couple additional sentences briefly and "birds eye view" explaining what constitutes adaptive will help orient readers. The easiest way to this – just move "Briefly,…each of these timepoints." to the Results section.

12) Jumping off of the last comment – why wasn't dN/dS done? Given that dN/dS is more commonly applied, I think a comparison of these results to standard dN/dS analysis is merited. In fact, including a dN/dS analysis may bolster the authors' overall conclusions and/or contribute to justifying using the Bhatt method, especially if dN/dS is not sufficient sensitive for this data.

13) In general, the authors should clarify their use of the terms "positive selection" and "adaptive substitutions." The former is traditionally associated with interpreting dN/dS, which isn't calculated in the manuscript, and the latter term is more mechanistically-oriented regarding effects of mutations. Therefore, what is meant by "positive selection" in the simulations, and how does this definition/implementation compare to the authors' measurements of "adaptive substitutions"?

14) Figure 7 and its associated analysis raised some concerns for me. It seems like only simulation 5 replicates were performed for each condition. Is there a reason so few simulations were performed (e.g. too computationally expensive?). Further, mean and CI bars for only 5 replicates in Figure 7 gives the impression that there are more than 5 replicates. A strip plot would be more forthcoming about the analyses conducted here, and some additional explanation about why only 5 replicates were performed per condition would help.

15) Table 1 and its associated analysis:

– CI's or some measure of statistical bounds should be included in Table 1.

– Where is OC43B in the table? Was the analysis not performed on this lineage, and if so why not?

– The authors motivate this analysis by explaining how TMRCA is meaningful for H3N2. Can the authors perform this analysis for H3N2 proteins as well to provide further context for the HCoV values, just as they did for these analyses associated with Figure 5?

16) Results, tenth paragraph. To motivate this analysis, the authors may also wish to cite this paper, co-authored by Trevor, that uses a downsampling strategy from empirical to study 2009 H1N1 dynamics, and shows time dependency in evolutionary metrics. https://bedford.io/papers/meyer-time-dependence/

– In addition, why did the authors use simulated data here? If we have HCoV sequence data since at least the 1990s, it seems possible to have used real data here. Further explanation/justification is therefore needed.

– All that said, looking at the CI's (assuming these are CI's – the legend needs to add this info) in Figure 7—figure supplement 1, the bounds across time points are often overlapping. One might expect that CIs would be wider as sample size decreases, which is not always the case. To my eye, the *only* panel in this figure that truly shows the authors' conclusion is the "no recombination/high positive selection" panel.

17) Please add a reference in the third paragraph of the Discussion about transmissibility/pathology correlates.

18) Subsection “Phylogenetic Inference”: IG-Tree typo should be IQ-TREE. In addition, The authors may also wish to confirm on their own end which IQ-TREE version was used – a major version 2 was released in 2020 and has a different citation. Either way, please indicate the IQ-TREE version used and make sure the citation is right for whichever was used.

19) Figure 2—figure supplement 1C – please choose different colors. Since some transparency is used for points, it's very hard to distinguish precisely light from dark purple.

20) Grammar and spelling:

– There should be a comma after 229E (as in, "…two species of HCoV, OC43 and 229E, were identified…")

– "Some human respiratory illness…" is a runon sentence. Please add a comma before ",while others,".

– Legend of Figure 1: Please add a comma at "…for each viral gene, and those…"

– Typo, "spikenand" → "spike and"

– Please add a comma before ", while the rate of adaptation…"

– Discussion, third paragraph, again a comma is needed before "while."

---

## [Author Response]

Reviewer #1:The authors use dN/dS-based statistics to show that the S1 domain has been adapting in coronaviruses OC43 and 229E, but not in NL63. For at least OC43 and 229E, this agrees qualitatively with earlier studies. The authors should make a detailed, quantitative comparison with these earlier results.1) I had a hard time understanding exactly what was going on in the simulations, and whether, e.g., they produced phylogenies that looked like the real ones. But perhaps the simulations should be cut or de-emphasized. From my understanding of the Bhatt et al. papers and the authors' code, their selection-detection method is entirely site-based and doesn't use the tree at all, and so should be insensitive to recombination a priori. If this is correct, the simulations are largely unnecessary (but see the next point).

We agree that Bhatt method should be robust to recombination since it is a site-based method. However, we think the simulated data in Figure 7 provides a useful confirmation of this in addition to providing a framework to test how limited longitudinal sequencing data affects the estimated rate of adaptation (Figure 7—figure supplement 1). In order to provide clarity, we have added Figure 7—figure supplement 2, which shows representative phylogenies for each of the recombination/selection. When presenting early iterations of this work to colleagues, we received repeated questions about potential bias from recombination and so although the Bhatt method is indeed robust to recombination (in theory and in practice), we feel it’s still useful to show this robustness explicitly.

2) It looks like there is substantial geographic population structure, as well as time-varying geographic biases in sampling. This seems like it could generate false signals of substitutions, where standing spatial variation appears to be temporal variation. Can the authors bound how large this contribution could be? If not, perhaps they could still say that they've found signatures of either global positive selection or local adaptation, both of which are interesting.

Examining the phylogenies with isolates colored by country

(https://nextstrain.org/community/blab/seasonal-cov-adaptive-evolution/oc43/spike?c=country) , we can see that the tree forms a ladder-like topology where the tips are organized by date, rather than by country. This is especially visible when examining sequences from the 2010’s, where isolates from several countries were sequenced. In these years, isolates frequently appear interspersed within the same temporal cluster rather than forming separate geographic clusters. This suggests that HCoVs are rapidly transmitted globally and this geographic mixing justifies analyzing all isolates together as one population, rather conducting separate analyses on geographically-partitioned data. We have added an explanation of this to the Results section, which reads as follows:

“Additionally, the trees form ladder-like topologies with isolate tips arranged into temporal clusters rather than geographic clusters, indicating a single global population rather than geographically-isolated populations of virus”.

3) Why is synonymous divergence also much higher in OC43 S1 than it is in RDRP or S2?

We are not sure why synonymous divergence differs between genes. To address this concern, we have calculated the ratio of nonsynonymous divergence in spike to nonsynonymous divergence in RdRp and compare this to the equivalent ratio of synonymous divergence. We also compute the same ratios for S1 versus S2. We find that the nonsynonymous divergence ratio is higher than the synonymous divergence ratio. This indicates that, while spike synonymous divergence is often higher than RdRp synonymous divergence, this difference is much smaller than the difference in nonsynonymous divergences. We take this to be a convincing result showing that nonsynonymous divergence in spike is indeed higher than nonsynonymous divergence in RdRp (and the same for the comparison of S1 to S2). We have added Figure 3—figure supplement 2 showing these results, and have inserted the following text in the Results section to acknowledge this observation:

“Though we would expect synonymous divergence to be equivalent in all areas of the genome, this is not born out in our results. […] Thus, despite differences in synonymous divergence, spike is accumulating more relatively more nonsynonymous divergence than RdRp.”

4) As I mention in the general assessment above, the authors should compare their results to previous studies. This includes both overall substitution rates as well as specific sites found to have repeated substitutions. For NL63, how does the null result compare to what's known? I believe Kiyuka et al., 2018, found widespread reinfection by similar NL63 genotypes, which may be relevant.

Thanks for pointing us towards the Kiyuka et al. paper – we have included a brief discussion of this paper in the Introduction and Discussion sections. Kiyuka et al. designate multiple NL63 genotypes and show that each genotype circulates within Kilifi Kenya for a relatively long period of time. Additionally, people become reinfected by the same genotype and Kiyuka et al. observe that NL63 reinfections are often enhanced by a previous infection. Kiyuka et al. hypothesize that enhanced reinfection signals that NL63 is exploiting an immune response and that epitopes would actually be under purifying selection because of this. Whether or not this hypothesis is correct, their observations of persistent circulation of NL63 genotypes and reinfection by the same genotype fit well with our results showing a lack of adaptive evolution in NL63 S1.

“Our finding fits with a study of NL63 in Kenya, which identified multiple genotypes of NL63 and show that people regularly become reinfected with the same genotype of NL63 (Kiyuka et al., 2018). […] Though our results cannot explain why OC43 and 229E likely evolve antigenically while NL63 does not, Kiyuka et al. observe that NL63 reinfections are sometimes enhanced by a previous infection and hypothesize that NL63 is actually under purifying selection at epitope sites (Kiyuka et al., 2018).”

We have also added a sentence in the Results comparing our calculation of dN/dS in OC43 spike to the estimates reported in Ren et al., 2015. The Chibo and Birch, 2006 paper states that “a probability of 1.0 was obtained when testing for positive selection”, but does not elaborate on specific dN/dS values.

“Our estimate of dN/dS in OC43 spike is similar to the previously reported estimate of roughly 0.3 (Ren et al., 2015).”

Reviewer #3:[…] Kistler and Bedford present a timely and highly relevant analysis of adaptive evolution in seasonal "common cold" coronaviruses. Overall, I find the research compelling, well-performed, and mostly well-presented (though some organizational changes are needed). As always, the commitment to open code and data from the Bedford lab is admirable and successfully performed/communicated. In my comments below, I offer advice to improve the clarity and presentation of the paper, with a few small requested analyses or need for further explanations.1) "We have arbitrarily labeled these lineages 'A' and 'B' (Figure 1)." The figure then shows A/B panels for different hCoV strains, which is a little confusing at first until you orient to the figure presentation. I recommend labeling the phylogeny in panel A with "A" and "B" rather than just using colors, and including that lineage information in the legend.

A legend showing the color for each lineage has been added to Figure 1 and the associated figure supplements.

2) The legend for Figure 1 needs units for the clock rate. Presumably this is the codon sub rate per year, which is used elsewhere? Or is this nucleotide? Similarly, units are also needed:– Subsection “Rate of Adaptation in RdRp and subdomains of spike”, for the parenthetical "(or 0.45 adaptive substitutions each year)". Are the authors converting to nucleotide?– Subsection “Phylogenetic Inference”.

The clock rate is given in substitutions per nucleotide site per year. These units have been added to the text. The parenthetical was altered for clarification and now reads: “(or 0.45 adaptive amino acid substitutions in S1 each year)”.

3) In general, I find the references to supplementary figures in the text confusing. For example, I first read the phrase "Figure 1 Supplement 1A" to mean both Figure 1 and Supplement 1A. Writing this as, "Figure 1—figure supplement 1A" will make it more clear that Figure 1 of the main text is not being referenced.

References to figures and supplements have been changed according to *eLife* convention, so that figures are referred to by (Figure 1) in the text and supplements are referred to by (Figure 1—figure supplement 1).

4) Results, second paragraph – sentences are not well ordered. Should be in order: 1) Though…, 2) Because…, 3) This….

We have restructured this paragraph to, hopefully, make the explanation of how we assigned lineages and completed gene-specific analyses more clear.

5) The last two sentences in the second paragraph of the Results seem tacked on and not immediately relevant to recombination. Please move these sentences or include a paragraph break.

We have introduced a paragraph break as well as changing the text in previous sentences to, hopefully, make this section more clear to the reader.

6) Related to the previous comment, the Results section as a whole will benefit from improved organization, specifically by creating subsections. I highly recommend adding these to improve readability.

We have added subsections to the Results section.

7) Comments for Figure 2:– Unless it becomes too busy, small indicators (or at least in the legend to avoid figure noise) might be added to emphasize the RBD domain within S1 specifically.

We have added the 229E RBD and the putative OC43 RBD to Figure 2. We have also added a sentence to the Results that explains that we conduct analyses on all of S1 rather than just the RBD within it because neutralizing antibodies have been identified which bind to the N-Terminal Domain (NTD) as well as the RBD.

– I recommend changing the color of the asterisks in panel A to match the lineage A (red) color.

We have changed the layout of this figure to put OC43 lineage A and lineage B in separate panels, which should reduce the confusion due to over-layed data. We have decided to keep the asterisks black so that is clear that they are asterisks rather than data points and hope that separating the 2 lineages into different panels will obviate the need for coloring the asterisks.

8) The authors write, "from that lineage's common ancestor." It would be more precise to say the "from that lineage's most recent common ancestor."

We agree and have changed this sentence to read “most recent common ancestor”.

9) Figure 3, specifically in panel A for RdRp, leaves some ambiguous interpretations: Is the line missing for OC43 lineage B because there is no RdRp data after the early 1990s (seems unlikely?), or because there were no adaptive substitutions, in which case the orange lines should remain steady at 0? This aspect of the figure should be clarified or fixed.

We have added a sentence in the Figure 3 legend to clarify this.

There is a similar situation for panel C, HKU1 lineage B, in Figure 2—figure supplement 1. In addition, the "C" for that panel is cut off at the bottom, so this figure needs to be slightly reformatted.

We have reformatted the supplement to avoid the panels being cut off.

10) In the fifth paragraph of the Results, the authors introduce the H3N2 analysis, but the actual analysis is not really presented or described for another two pages. The H3N2 comparison is definitely not part of Figure 4, which this paragraph is introducing. I think this sentence is likely misplaced? Again, this comment shows that adding subsections to the Results section will be helpful for overall organization.

We have removed the mention of H3N2 in the Introduction of the Bhatt analysis. We now mention calculating rates of adaptation in H3N2, measles and flu B several paragraphs later, when we actually compare these rates to seasonal HCoVs.

11) Results, I would like to see more details about what constitutes an “adaptive substitution” in the Bhatt method, which is not as widely used as dN/dS, within the main text itself. A couple additional sentences briefly and "birds eye view" explaining what constitutes adaptive will help orient readers. The easiest way to this – just move "Briefly,…each of these timepoints." to the Results section.

We agree that it would be useful to orient the reader to the Bhatt method by providing a brief overview of the method when it is introduced in the Results section. We have included this explanation in the Results section as suggested and rearranged the Methods description of the Bhatt method accordingly.

“Briefly, this method defines a class of neutrally-evolving nucleotide sites as those with synonymous mutations or where nonsynonymous polymorphisms occur at medium frequency. […] This method compares nucleotide sequences at each timepoint (the ingroup) to the consensus nucleotide sequence at the first time point (the outgroup) and yields an estimate of the number of adaptive substitutions within a given genomic region at each of these timepoints.”

12) Jumping off of the last comment – why wasn't dN/dS done? Given that dN/dS is more commonly applied, I think a comparison of these results to standard dN/dS analysis is merited. In fact, including a dN/dS analysis may bolster the authors' overall conclusions and/or contribute to justifying using the Bhatt method, especially if dN/dS is not sufficient sensitive for this data.

We originally only included an analysis of the data using the Bhatt method because this method (and implementations of the McDonald-Krietman method in general) is better at distinguishing relaxed selection from positive selection, whereas dN/dS can get confused in these situations. However, while we think dN/dS is less appropriate for detecting selection in RNA virus populations, we agree that including this standard method for comparison is useful. We have included dN/dS estimates for each HCoV and gene in Table 1. As expected, here we see higher dN/dS values for spike than RdRp and higher dN/dS values for S1 than S2.

13) In general, the authors should clarify their use of the terms "positive selection" and "adaptive substitutions." The former is traditionally associated with interpreting dN/dS, which isn't calculated in the manuscript, and the latter term is more mechanistically-oriented regarding effects of mutations. Therefore, what is meant by "positive selection" in the simulations, and how does this definition/implementation compare to the authors' measurements of "adaptive substitutions"?

While the terms “positive selection” and “adaptive evolution” are not synonymous, they are related in that adaptive evolution is a result of positive selection acting on genetic variation. We have added a sentence to the Results section to explain this and aim to use these terms in throughout our paper as they are used in the literature (i.e. “positive selection” to refer to dN/dS results and “adaptive substitutions” to refer to the Bhatt method).

“Mutations that escape from population immunity are beneficial to the virus and so are driven to fixation by positive selection. This results in adaptive evolution of the virus population.”

For the simulations, we specify a fitness function that imitates positive selection at epitope sites, then we use the Bhatt method to detect adaptive evolution, the result of this positive selection. We have added more detail to the Materials and methods section describing how these simulations are done.

“Positive selection was simulated through exposure-dependent selection at a subset of spike S1 sites proportional to the number of epitope sites in H3N2 HA (Luksza and Lässig, 2014). The simulated selection allows mutations in these “epitope” sites to rise in frequency while also encouraging “epitopes” to change over time (to mimic antigenic novelty).”

14) Figure 7 and its associated analysis raised some concerns for me. It seems like only simulation 5 replicates were performed for each condition. Is there a reason so few simulations were performed (e.g. too computationally expensive?). Further, mean and CI bars for only 5 replicates in Figure 7 gives the impression that there are more than 5 replicates. A strip plot would be more forthcoming about the analyses conducted here, and some additional explanation about why only 5 replicates were performed per condition would help.

We have run additional simulations so Figure 7 now displays mean and 95% CI for 10 replicates instead of 5.

15) Table 1 and its associated analysis:– CI's or some measure of statistical bounds should be included in Table 1.

We have added 95% confidence intervals to the table of TMRCA values (now Table 2).

– Where is OC43B in the table? Was the analysis not performed on this lineage, and if so why not?

We did not perform this analysis on OC43 lineage B because the limited number of RdRp sequences available for this lineage could artificially skew the TMRCA estimates. We have added a sentence to the Results section stating this.

We did not perform this analysis on OC43 lineage B because the limited number of RdRp sequences available for this lineage could artificially skew the TMRCA estimates. We have added a sentence to the Results section stating this.

– The authors motivate this analysis by explaining how TMRCA is meaningful for H3N2. Can the authors perform this analysis for H3N2 proteins as well to provide further context for the HCoV values, just as they did for these analyses associated with Figure 5?

Based on Bedford et al., 2011, our estimates of HCoV TMRCA are roughly 2-2.5 times longer than TMRCA of H3N2 hemagglutinin. We have included this information in the Results section.

16) Results, tenth paragraph. To motivate this analysis, the authors may also wish to cite this paper, co-authored by Trevor, that uses a downsampling strategy from empirical to study 2009 H1N1 dynamics, and shows time dependency in evolutionary metrics. https://bedford.io/papers/meyer-time-dependence/– In addition, why did the authors use simulated data here? If we have HCoV sequence data since at least the 1990s, it seems possible to have used real data here. Further explanation/justification is therefore needed.

While we could use down-sampling strategy of the empirical data to show a decrease in power with a decrease in data, we think that using simulated data to show this relationship is better because it gives a “truth” to compare to. In other words, in the simulated data we know whether or not there is positive selection on residues within S1 and, therefore, whether or not adaptive substitutions should be detected. Using *all*  of the simulated data gives us a “true” rate of adaptation that we can compare other rates of adaptation to. Empirical HCoV data is already somewhat sparse and, therefore, we do not think the rates of adaptation we calculate can be regarded as a “truth”, but rather are estimates.

– All that said, looking at the CI's (assuming these are CI's – the legend needs to add this info) in Figure 7—figure supplement 1, the bounds across time points are often overlapping. One might expect that CIs would be wider as sample size decreases, which is not always the case. To my eye, the only panel in this figure that truly shows the authors' conclusion is the "no recombination/high positive selection" panel.

We have added clarification to the Figure 7—figure supplement 1 to state that the mean and 95% CI are plotted for each point. Additionally, we ran more simulations (for a total of 10). Our reasoning in including this figure supplement is to show that when the 30-year point is taken as the “true” rate of adaptation, it is evident that a reduced amount of temporal data alters the estimated rate of adaptation. Though the general trend is that less longitudinal sequence data results in a lower estimated rate of adaptation and higher uncertainty, this is not the case at every time point. However, whether a truncated time-series results in a lower or higher rate of adaptation, the estimated rate is still skewed from the “true” rate. We have added a sentence to the Results section to try to clarify this interpretation of Figure 7—figure supplement 1.

17) Please add a reference in the third paragraph of the Discussion about transmissibility/pathology correlates.

We have rewritten this sentence to remove this statement.

18) Subsection “Phylogenetic Inference”: IG-Tree typo should be IQ-TREE. In addition, The authors may also wish to confirm on their own end which IQ-TREE version was used – a major version 2 was released in 2020 and has a different citation. Either way, please indicate the IQ-TREE version used and make sure the citation is right for whichever was used.

The typo in the Materials and methods section has been fixed, the version number was added, and the citation was verified.

19) Figure 2—figure supplement 1C – please choose different colors. Since some transparency is used for points, it's very hard to distinguish precisely light from dark purple.

The typo in the Materials and methods section has been fixed, the version number was added, and the citation was verified.

20) Grammar and spelling:– There should be a comma after 229E (as in, "…two species of HCoV, OC43 and 229E, were identified…")

Done.

– "Some human respiratory illness…" is a runon sentence. Please add a comma before ",while others,".

Done.

– Legend of Figure 1: Please add a comma at "…for each viral gene, and those…"

Done.

– Typo, "spikenand" → "spike and"

Addressed.

– Please add a comma before ", while the rate of adaptation…"

Done.

– Discussion, third paragraph, again a comma is needed before "while."

Fixed.